# A New Optimal Subset Selection Method of Partial Ambiguity Resolution for Precise Point Positioning

Caiya Yue [1,2,*] , Yamin Dang [3], Shuqiang Xue [3], Hu Wang [3], Shouzhou Gu [3] and Changhui Xu [3]

1 School of Geography and Environment, Liaocheng University, Liaocheng 252059, China
2 Beijing Key Laboratory of Urban Spatial Information Engineering, Beijing 100045, China
3 Chinese Academy of Surveying and Mapping, Beijing 100830, China
* Correspondence: yuecaiya@lcu.edu.cn; Tel.: +86-184-3933-7506

**Abstract:** Rapid and accurate ambiguity resolution is the core of high-precision precise point positioning (PPP) data processing. However, the ambiguity parameters in PPP observation models are easily affected by atmospheric residual and gross errors, which lead to the probability of successfully fixing decreases and computational burden increases in full ambiguity resolution. Therefore, an increasing number of partial ambiguity resolution (PAR) strategies have been proposed. The selection of the optimal subset of PAR is crucial in this method. The traditional optimal subset selection method of PAR commonly leads to a single judgment criterion and weakened geometric configuration strength because the satellites with low elevation angles are often easily eliminated during the optimal subset selection. In this paper, a multi-factor constrained optimal subset selection method for PAR was proposed, which incorporates the ambiguity variance, the ambiguity dilution of precision (ADOP), satellite position dilution of precision (PDOP) and ratio test values. In order to verify the feasibility of the proposed optimal subset selection method, PAR tests under two schemes were performed for GPS/Galileo based on the static observation data of 15 Multi-GNSS Experiment (MGEX) tracking stations. The results show that, compared with the ambiguity variance sorting method, the proposed subset selection method can further improve the accuracy of the coordinate solution and the strength of geometric figure positioning. The average root mean square of the coordinate residuals is found to decrease by about 12.90%, 6.83% and 9.39% in the eastern, northern and vertical directions, respectively. The increase in the fixed epoch rate ranged from 0.87% to 33.33%, with an average of about 8.71%.

**Keywords:** partial ambiguity resolution; optimal subset selection; precise point positioning; static observation



## 1. Introduction

Precise point positioning (PPP) can achieve high-accuracy position information in any range of the world flexibly and efficiently without setting up base stations [1]. Thus, PPP is considered as a new precision positioning mode following real-time kinematic (RTK) and network RTK positioning. With the continuous development of the global navigation satellite system (GNSS), multi-frequency, multi-system joint positioning has enhanced multi-level applications for PPP, such as the analysis of ionospheric refraction effects [2], real-time retrieval of precipitable water vapor [3], inversion of earthquakes and crustal deformation [4], unmanned driving in the urban environment [5] and integrity monitoring [6].

Although the PPP technique can bring great advantages, whether the ambiguity on the carrier phase observations can be resolved rapidly and accurately has been a key issue limiting its further application. For ionosphere-free combination, UofC combination, undifferenced uncombined, single-system, multi-system, dual-frequency and multi-frequency, all require a period of convergence to achieve a high-precision PPP solution [7–9]. The

main reason is that the phase hardware delay bias at the satellite and receiver ends is highly linearly correlated with the non-differential ambiguity. This results in the loss of the whole-cycle characteristic of solved ambiguity parameters, the fractional part of which is called fractional cycle bias (FCB) [10]. This type of bias is difficult to be eliminated at the satellite side and receiver ends, contrary to the bias in the dual-difference observation model. For PPP, FCB is estimated mainly based on the ground network and then corrected at the user end. Gabor et al. firstly proposed to extract FCB at the satellite end using the inter-satellite single-difference method, thus achieving wide-lane and narrow-lane ambiguity resolution [11]. However, the narrow-lane FCB estimation and the narrow-lane ambiguity resolution could not be achieved due to the limited accuracy of the precise orbit and clock products. With the increasing accuracy of ephemeris products released by the International GNSS Service (IGS), FCB separation methods, decoupled satellite clock and integer-recovery clock have been proposed to address the satellite-end FCB [12–14]. Then, the ambiguity-fixed PPP solution can be achieved. The results show that the performance of static single-day solutions can be further improved, especially in the east direction where the accuracy is more significantly enhanced. Since then, almost all strategies for FCB processing and ambiguity-fixed PPP solutions of GNSS have been developed or improved based on these three methods. Several institutions have been able to broadcast real-time or post-processing FCB products for ambiguity resolution. In summary, the processing methods and release models of the FCB products required for ambiguity-fixed PPP solutions have been well established [15–17].

In GNSS PPP, the accuracy and reliability of the parameter solution are theoretically higher when more ambiguities are correctly fixed. However, for PPP solving models, there are more parameters to be estimated. The correlation between parameters is strong, and thus most of the common errors cannot be eliminated by forming dual-difference observations. This results in relatively low ambiguity accuracy and difficulties in achieving full ambiguity resolution [18]. In addition, full ambiguity resolution will increase the computational burden, especially in multi-system, multi-frequency uncombined PPP. Therefore, Teunissen et al. [19] proposed the partial ambiguity resolution (PAR) method. Many studies have developed various improved PRA methods to improve the resolution performance and increase the ambiguity resolution rate. The core technique of PAR is the selection of the optimal ambiguity subset. Odijk and Teunissen proposed an ambiguity dilution of the precision (ADOP) ranking method, which determined the subplot of ambiguities with higher accuracy based on the accuracy of each float ambiguity and their correlations [20]. Takasu and Yasudada et al. considered that satellite observations were subjected to more multi-path effects and atmospheric delay errors at a lower elevation angle and proposed an elevation-angle-based ambiguity resolution method. The method is as follows: the float ambiguity is sorted according to the elevation angle in descending order; then, the satellites with smaller elevation angles are removed when the ambiguity is attempted to be resolved until it passes the ambiguity check [21]. Homoplastically, a sorting method according to the variance size was proposed to realize ambiguity resolution based on the original ambiguity accuracy (the Var-sort method) [22]. Parkins proposed a signal-to-noise ratio (SNR) sorting method based on the ratio of received carrier signal strength-to-noise strength [23]. The core of this method is that when the full ambiguity resolution fails, the satellite ambiguity subset with a larger SNR will be preferentially resolved. Although the elevation-angle-based method, the SNR method, the ADOP method and the Var-sort method can improve the efficiency of ambiguity resolution and enhance the coordinate solution, they have a common disadvantage. When the ambiguity subset is selected, the satellites with low satellite elevation angles are commonly eliminated, resulting in a relatively weak satellite geometric strength. This is unfavorable to obtaining high-precision parameter solutions in some regions with serious occlusion environments. Then, Wang and Feng proposed a method of selecting the optimal subset from decorrelated ambiguity vectors [24]. Their validation results show that the proposed method was beneficial to select ambiguity subsets with sufficiently high-resolution rates in a multi-system observation environment. This

method is also more reliable compared with full ambiguity resolution. Li et al. (2016) simultaneously used the bootstrapping success rate of the fixed solution and the ratio test to check the full and partial ambiguity subsets; then, the combined satellite pair with the highest variance after the linear ambiguity combination is eliminated sequentially until the optimal solution set of linear combinations of ambiguity is obtained [25]. This method has been applied to the ambiguity resolution of GPS ionosphere-free combinations and BDS triple-frequency uncombined PPP [16,26]. However, the key to partial ambiguity resolution is to select the required optimal subset considering ambiguity accuracy, satellite geometric configuration strength and computational efficiency.

In this study, a multi-factor constrained optimal subset selection method of PAR is proposed. This method integrates ambiguity variance, ADOP, position dilution of precision (PDOP) and ratio-test values and avoids the shortcoming that the satellites with smaller elevation angles are always discarded and weak positioning geometry strength is induced in the traditional method. The performance of the proposed method is verified using GPS/Galileo observations of 15 Multi-GNSS Experiment (MGEX) continuous tracking stations.

## 2. Methods

The PAR model was firstly derived by combining the least-squares ambiguity decorrelation adjustment (LAMBDA) algorithm. Then, an integrated multi-factor constrained optimal subset selection method of the PAR was proposed.

### 2.1. PPP Partial Ambiguity Resolution Model

In the solving process of uncombined PPP, the ionospheric delay deviation is linearly correlated with the float ambiguity, and the narrow-lane ambiguity of each satellite (pair) is also highly correlated [26]. Compared with the original carrier wavelengths, the combined wide-lane wavelengths are longer and can be resolved by the rounding algorithm. The narrow-lane ambiguity can be obtained through the LAMBDA search. In essence, the LAMBDA method is a least squares-based search algorithm for integer solutions [25]. The carrier phase observation equation in uncombined PPP can be simplified as

$$y = Bb + Aa + \varepsilon \tag{1}$$

where $y$ is the GNSS observation vector; $a$ is the integer ambiguity parameter, $a \in Z^n$; $b$ is the parameter vector containing station position and atmospheric delay, $b \in R^p$; $\varepsilon$ is the observation noise; and $A$ and $B$ are the design matrices of the corresponding parameters, respectively.

According to the nature of the parameters to be estimated, Equation (1) is a mixed-integer least squares estimation. By neglecting the integer ambiguity constraint, Equation (2) can be obtained based on the least-squares principle:

$$\min_{a,b} \|y - Bb - Aa\|^2_{Q_y^{-1}} \tag{2}$$

where $\|\bullet\|^2_{Q_y}$ is the weighted squared norm. After the equation is solved, the station location information and atmospheric delay parameters can be expressed as $\hat{b}$, and the float ambiguity solution can be expressed as $\hat{a}$. The corresponding covariance matrix can be expressed as

$$\begin{bmatrix} \hat{b} \\ \hat{a} \end{bmatrix} Q_y = \begin{bmatrix} Q_{\hat{b}} & Q_{\hat{b}\hat{a}} \\ Q_{\hat{b}\hat{a}} & Q_{\hat{a}} \end{bmatrix} \tag{3}$$

Based on the integer property of ambiguity parameters, the smallest integer vector satisfying the following objective function (Equation (4)) is the integer ambiguity solution.

$$\min_a \|\hat{a} - \widehat{a}\|^2_{Q_a^{-1}} \tag{4}$$

After the float ambiguity is resolved to obtain the integer ambiguity, the unknown parameters $b$ and the variance–covariance array $Q_b$ can be updated using Equation (5) for parameter estimation in the next epoch.

$$\begin{cases} \widehat{b} = \hat{b} - Q_{\hat{b}\hat{a}}Q_{\hat{a}}^{-1}(\widehat{a} - \hat{a}) \\ Q_{\widehat{b}} = Q_{\hat{b}} - Q_{\hat{b}\hat{a}}Q_{\hat{a}}^{-1}Q_{\hat{b}\hat{a}} \end{cases} \qquad (5)$$

where $\widehat{b}$ and $Q_{\widehat{b}}$ are the updated parameter solution and the variance–covariance array, respectively. The key is to continuously search for the float ambiguity to obtain the least-squares solution satisfying Equation (2). Currently, the most theoretically rigorous and efficient ambiguity resolution algorithm is the LAMBDA method proposed by Teunissen et al. [19]. The core of the LAMBDA algorithm consists of two parts: the ambiguity decorrelation based on the integer transform and the integer ambiguity search based on the sequential conditional least-squares estimation [25].

However, for uncombined PPP models, their parameters have strong correlations. The float ambiguity parameters are also susceptible to unmodeled errors (e.g., atmospheric residual errors) and other gross errors. In addition, the ambiguity search space is gradually enlarged as the ambiguity number increases. These factors lead to the tendency to increase the time for the first ambiguity resolution and decrease the resolution probability in full ambiguity resolution. Therefore, PAR methods are introduced to improve the reliability and performance of ambiguity resolution.

It is assumed that $\hat{a} = (\hat{a}_1, \hat{a}_2, \cdots\cdots, \hat{a}_n)$ is the float ambiguity vector in an epoch; $n$ is the number of float ambiguity parameters; $\hat{a}_m = (\hat{a}_i, \hat{a}_{i+1}, \cdots\cdots, \hat{a}_m)$ is the subset component of $\hat{a}$, i.e., $\hat{a}_m \in \hat{a}$; and $m$ is the number of float ambiguities in the subset component ($m < n$). If $\hat{a}_m$ is the optimal subset vector required for PAR in the current epoch, the variance–covariance matrix $Q_{\hat{a}}$ of full-float ambiguity parameters can be decomposed as

$$Q_{\hat{a}} = \begin{bmatrix} Q_{\hat{a}_m} & Q_{\hat{a}_{(m)(n-m)}} \\ Q_{\hat{a}_{(n-m)(m)}} & Q_{\hat{a}_{n-m}} \end{bmatrix} \qquad (6)$$

The LAMBDA method can be applied to the selected vector subset $\hat{a}_m$ and the corresponding covariance array $Q_{\hat{a}_m}$ to search for the integer least-squares solution. When the ambiguity in the optimal subset is resolved (assumed to be $\widehat{a}_m$), it can be substituted into Equation (5) to determine the remaining ambiguity $\widehat{a}_{n-m}$, the position parameter $\widehat{b}$ and the variance–covariance array $Q_{\widehat{b}}$.

$$\begin{cases} \widehat{b}_{|\widehat{a}_m} = \hat{b} - Q_{\hat{b}\hat{a}_m}Q_{\hat{a}_m}^{-1}(\hat{a}_m - \widehat{a}_m) \\ \widehat{a}_{(n-m)|\widehat{a}_m} = \hat{a}_{(n-m)} - Q_{\hat{b}\hat{a}_m}Q_{\hat{a}_m}^{-1}(\hat{a}_m - \widehat{a}_m) \\ Q_{\widehat{b}} = Q_{\hat{b}} - Q_{\hat{b}\hat{a}_m}Q_{\hat{a}_m}^{-1}Q_{\hat{b}\hat{a}_m} \end{cases} \qquad (7)$$

### 2.2. New Subset Selection Method for PAR

The above analysis clearly shows that the key to PAR lies in selecting the optimal ambiguity subset. In this paper, a multi-factor constrained optimal subset selection method is proposed. This method incorporates ambiguity variance, ADOP, PDOP and ratio-test values, and thus can solve the problem that only a single judgment criterion is usually adopted in the traditional optimal subset selection of partial ambiguity. In the presented method of this paper, the ambiguity variance is one of the important indicators of parameter estimation accuracy and can be extracted from the variance–covariance array after Kalman filtering. ADOP can describe the average accuracy of ambiguity parameters in the selected

subset and is expressed as Equation (8). PDOP is an important indicator of measuring the satellite-station geometric strength and is expressed as Equation (9).

$$ADOP = \sqrt{\det(Q_{\hat{a}})}^{\frac{1}{m}} \tag{8}$$

$$PDOP = \sqrt{trace\_p(A^{\mathrm{T}}A)^{-1}} \tag{9}$$

where $Q_{\hat{a}}$ is the variance–covariance array of the selected subset; the meaning of *m* is the same as that found in Equation (6); *A* is the design matrix of the positioning equation for the subset; and *trace_p* is the position variance element in the extracted matrix.

The detailed procedures of the optimal subset selection method of PAR are as follows:

(1) The float ambiguity and variance–covariance information of all satellites (pairs) are substituted into the LAMBDA algorithm to be searched and resolved. The resolved ambiguity is tested for the success rate and ratio test. If it passes the test, the full ambiguity resolution is performed; otherwise, the PAR is executed in the following steps.

(2) The variance of float ambiguity for all satellites (pairs) is sorted in ascending order.

(3) It is assumed that there is *n* ambiguity to be resolved in the current epoch. The last s (s < n) ambiguities with larger variance are selected for enumeration according to the number of tracked satellites and the complexity of observation conditions, i.e., $C_s^1, C_s^2, \cdots, C_s^r \cdots, C_s^s$ (where the *C* represents the permutation combination algorithm). There is a total of $\sum_{r=1}^{s} \frac{s!}{r!(s-r)!}$ small subsets. It is worth noting that the complex condition mainly refers to the degree of signal occlusion and multi-path interference around the GNSS station.

(4) Each small subset is combined with the previous $n - s$ ambiguity sequentially to obtain multiple candidate subsets *m*. Then, the LAMBDA search and ratio test are performed for each candidate subset, and ADOP and PDOP are calculated. To show common features with ADOP and PDOP, the inverse of the ratio-test value is computed. So far, multiple optimal subsets of candidates have emerged. The next steps are how to identify the optimal subset based on the observed environment.

(5) The inverse of the ratio-test value, ADOP and PDOP values of all combinations are normalized to a dimensionless quantity between 0 and 1, respectively.

(6) The resolution efficiency indexes $F_i$ of various combinations are assessed by introducing the weighting factors (*c_r*, *c_a* and *c_p*) of ratio, ADOP and PDOP, respectively, as shown in Equation (10). $\bar{r}_i$, $\overline{A}_i$ and $\overline{P}_i$ are the corresponding dimensionless quantities of the $i^{th}$ combination, respectively.

$$\begin{cases} F_i = c\_r \cdot \bar{r}_i + c\_a \cdot \overline{A}_i + c\_p \cdot \overline{P}_i \\ 1 = c\_r + c\_a + c\_p \end{cases} \tag{10}$$

(7) The subset corresponding to the minimum element in $F_i$ is taken as the optimal ambiguity subset.

The key to this new optimal ambiguity subset selection method is to determine the number (*m*) of ambiguities to be enumerated and three weighting factors (*c_r*, *c_a* and *c_p*). An excess *m* will increase the computational burden, while insufficient *m* will make it difficult to enumerate the optimal ambiguity subsets. For the GPS/Galileo combination, statistics show that about 10.2 satellites can be used for narrow-lane ambiguity resolution. Therefore, *m* was set as 4 in this study. In a continuous tracking station with good observation conditions, the weighting factor characterizing PDOP can be reduced, while the factors characterizing ADOP and ratio test can be enlarged. In urban and ravine environments with severe satellite obscuration, the geometric configuration strength of the satellite has a significant impact on positioning accuracy. Thus, the weighting factor of PDOP can be significantly enlarged and the weighting factor of ADOP can be increased properly, while the weighting factor of the ratio test needs to be reduced.

## 3. Results

In order to effectively evaluate the feasibility of the proposed optimal subset selection method for PAR, 15 MGEX continuous tracking station observations with a time sampling rate of 30 s for DOY 348 in 2019 were selected. The location of these stations is shown in Figure 1. The dual-frequency, uncombined static PPP float solution and ambiguity resolution with GPS/Galileo fusion were performed. The satellite elevation angle was set as 7° for the data solution and 15° for ambiguity resolution. The method of selecting the optimal subset according to the variance sorting method (i.e., the Var-sort method) was chosen for ambiguity resolution to compare with the proposed optimal subset selection method of partial ambiguity (i.e., the new method). It should be noted that the full ambiguity resolution was firstly performed for all satellites. Then, the partial ambiguity resolution was performed if it failed. The precise orbit and clock products were the post-processing products released by the IGS Analysis Center of Wuhan University, China. The FCB products used for ambiguity resolution were estimated according to the method proposed in the reference [26].

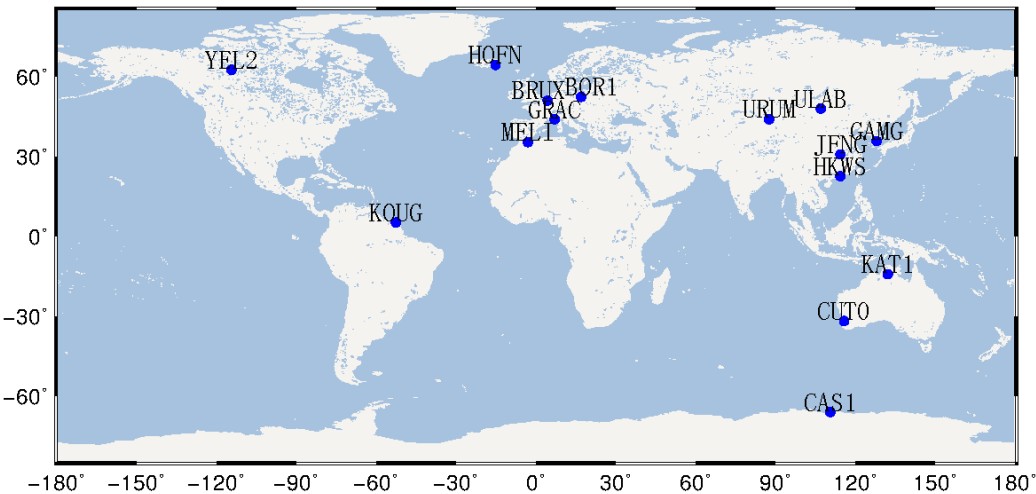

**Figure 1.** Information of MGEX continuous tracking stations.

### 3.1. Comparison of Float Solutions with Fixed Solution Using the Var-Sort Method

In order to verify the availability of test data and the reliability of the ambiguity resolution model, the performance of the PPP float solution and the fixed solution using the Var-sort method was first analyzed. The GPS/Galileo dual-frequency, uncombined PPP float solution and fixed solutions were obtained by dividing the 15 stations into arc segments every two hours. In Figure 2, BOR1 was taken as an example to demonstrate the time series of coordinate residuals of GPS/Galileo 2 h float and fixed solutions. The weekly solution coordinates published by IGS were selected as the coordinate truth values (https://www.igs.org/products, accessed on 30 December 2019). The analysis shows that when the ambiguity was resolved, the uncombined PPP solution improved in both convergence time and positioning accuracy.

In order to quantitatively analyze the performance of the fixed PPP solution, the coordinate residuals and convergence times of all stations per 2 h arc were calculated in this paper, respectively (as shown in Table 1). The accuracy of the coordinate solution for each 2 h arc was defined as the average of the absolute value of the coordinate residuals of all stations in the last 15 min. The convergence time was defined as the time required for the coordinate residuals to converge within 10 cm and remain stable in the subsequent epochs. The overall analysis shows that the positioning accuracy of both float and fixed solutions can be better than 2.0 cm in the horizontal direction and 3.0 cm in the vertical direction when the PPP solution was fully converged. It is found that the average positioning residuals of the GPS/Galileo float solution were about 1.82, 1.29 and 2.44 cm in the east, north and

up directions, respectively. When the ambiguity was resolved, the average residuals of the coordinates were about 1.13, 0.99 and 2.04 cm in the east, north and up directions, respectively. Compared with float solutions, static dual-frequency, uncombined PPP was improved by about 30.58% and 20.49% in the horizontal and vertical directions, respectively. As shown in Table 1, the time for coordinate residuals to converge within 10 cm was 21.2 and 13.4 min using the PPP float and fixed solution modes, respectively. The convergence time was improved by about 36.79%. The above results demonstrate the usability of the PPP ambiguity resolution model and the reliability of the precise orbit, clock difference and FCB products. They can be used to verify the effectiveness of the proposed subset selection method of partial ambiguity.

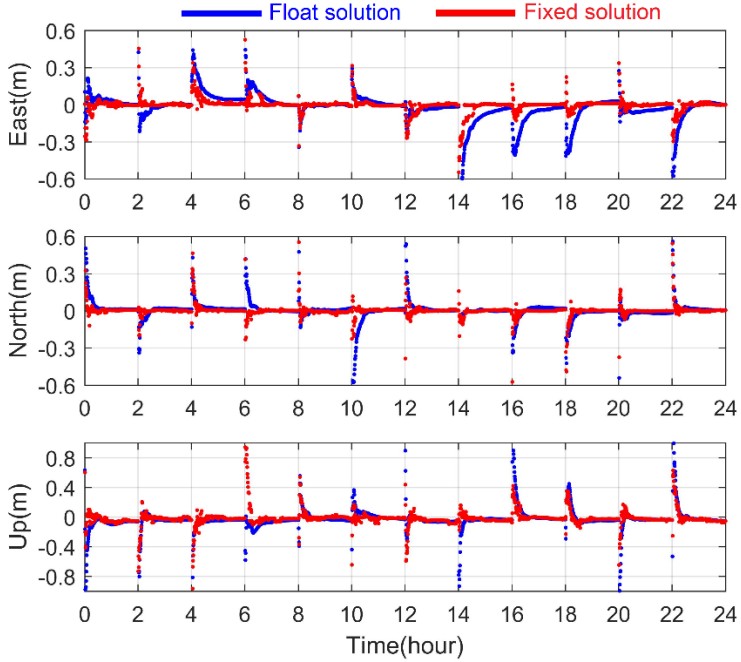

**Figure 2.** Coordinate residual series of float and fixed solutions at BOR1 station.

**Table 1.** Coordinate residuals and convergence time of uncombined PPP.

| PPP Mode | East Direction /cm | North Direction /cm | Up Direction /cm | Convergence Time /min |
|---|---|---|---|---|
| Float G/E | 1.82 | 1.29 | 2.44 | 21.2 |
| AR G/E | 1.13 | 0.99 | 1.94 | 13.4 |

### 3.2. Ambiguity Resolution Using the Proposed Method and the Var-Sort Method

The main difference between the PAR of the new method and the Var-sort method is the selection of the optimal subset. Due to its long wavelength, the wide-lane ambiguity can be easily resolved. Then, the resolution strategy used for both methods is the same. MGEX tracking stations are generally located in places with a good observation environment and the PDOP values do not change significantly during the 24 h observation period. Thus, the weight of PDOP in the scheme was set as the minimum. The ADOP has a relatively larger effect on ambiguity resolution in an open environment. Therefore, the weighting factors ($c_r$, $c_a$ and $c_p$) in this study were set as 0.3, 0.5 and 0.2, respectively.

The histograms of the coordinate residual distributions in the east, north and up directions for the two optimal subset selection schemes of partial ambiguity are presented in Figure 3. The weekly solution coordinates published by IGS were selected as the station reference coordinates. The "Continuous" strategy was adopted for ambiguity resolution, i.e., the float solution of the ambiguity estimated in the current epoch was used as the initial

value of the ambiguity in the next epoch. In order to visually analyze the performance, only the epochs that satisfied both the new method and the Var-sort method were extracted. The coordinate residuals of the two methods were counted based on the extracted epochs, with a total of 6205 data points. The confidence interval was assumed as 90%. The analysis shows that for both methods, with ambiguity resolution, the absolute value of coordinate residuals was better than 1.5 and 3.0 cm in the horizontal and up directions, respectively. The new subset selection method of partial ambiguity can further improve the accuracy of coordinate solutions compared with the Var-sort method. The average root mean squares (RMS) of coordinate residuals in the east, north and up directions were 0.66, 0.58 and 1.38 cm using the Var-sort method, and 0.57, 0.54 and 1.25 cm using the proposed method, respectively. The positioning accuracies in the horizontal, elevation and vertical directions were improved by about 12.90%, 6.83% and 9.39%, respectively.

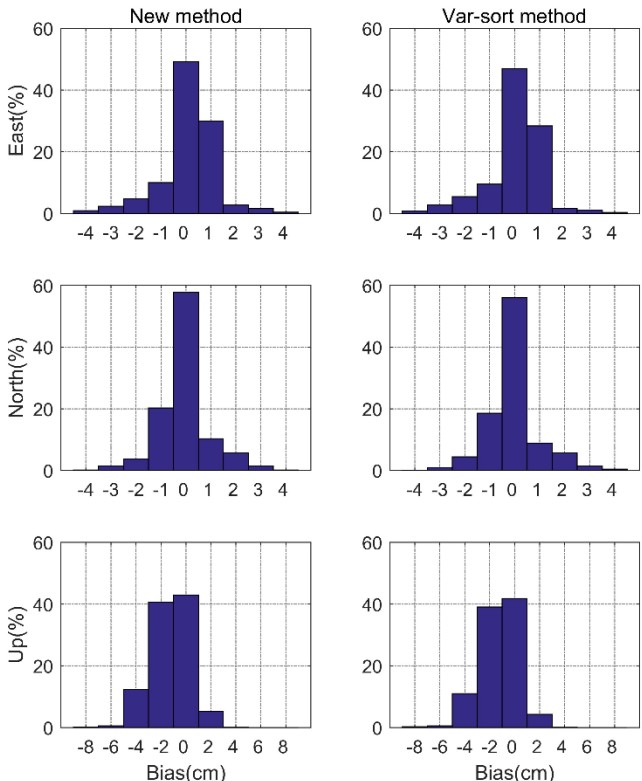

**Figure 3.** Histogram of coordinate residual distribution for different optimal subset selection methods for PAR.

In order to compare the ambiguity resolution enhancement rates using the two optimal subset selection methods, the number of epochs with a successful partial ambiguity resolution for all test stations was counted, as shown in Table 2. The results show that the number of epochs with ambiguity resolution can be further increased using the proposed method. Compared with the Var-sort method, the optimal subset selection method based on the factor constraint method can increase the number of epochs by 4~105, with an average increase of about 40.27 epochs (at a sampling interval of 30 s). The epoch with partial ambiguity increased by 0.87%~33.33%, with an average increase of about 8.71%. Particularly, the BRUX station had the highest increase in epochs by about 33.33%, while the CAS1 station had the smallest increase by about 0.66%. In order to explain this phenomenon, the number of satellites before the narrow-lane ambiguity resolution at BRUX and CAS1 stations was counted. The counted satellites meet various conditions, such as completing the wide-lane ambiguity resolution, passing the narrow-lane ambiguity resolution quality check and approaching the integer cycle threshold of the narrow-lane ambiguity resolution, and the subsequent narrow-lane ambiguity can be successfully resolved, as shown in Figure 4.

Statistics show that the average number of satellites at BRUX and CAS1 stations were about 8.68 and 12.11, respectively. If the minimum number of satellites required for narrow-lane ambiguity resolution was set as 4, then the number of satellites that can be adjusted at BRUX and CAS1 stations were 4 and 8, respectively. This indicates that the CAS1 station had more candidate subsets for the optimal subset selection of partial ambiguity. Thus, the success rate of ambiguity resolution with the traditional optimal ambiguity subset selection method was also higher at the CAS1 station, and the resolution rate of all epochs can reach 94%. In addition, the variation characteristics of the number of satellites show that the observation quality of the BRUX station was lower than that of the CAS1 station. However, the improved efficiency was more significant using the proposed method for partial ambiguity resolution. This indicates that the proposed optimal subset selection method of partial ambiguity offers a more significant improvement in the scenarios with poor observation quality or complex observation environments. In conclusion, the proposed multi-factor constrained optimal subset selection method can further improve the PPP solution in terms of ambiguity resolution rate and coordinate solution accuracy by evaluating the coordinate residuals and ambiguity resolution rate.

**Table 2.** Number of epochs of partial ambiguity resolution for each station under two optimal subset selection schemes.

| Station | Number of Resolved Epochs | | Difference in the Number of Resolved Epochs | Increased Epoch Resolution Rate (%) |
|---|---|---|---|---|
| | Proposed Method | Var-Sort Method | | |
| BOR1 | 279 | 243 | 36 | 12.90 |
| BRUX | 126 | 84 | 42 | 33.33 |
| CAS1 | 762 | 757 | 5 | 0.66 |
| CUT0 | 910 | 831 | 79 | 8.68 |
| GAMG | 460 | 456 | 4 | 0.87 |
| GRAC | 612 | 535 | 77 | 12.58 |
| HKWS | 852 | 824 | 28 | 3.29 |
| HOFN | 509 | 457 | 52 | 10.22 |
| JFNG | 806 | 763 | 43 | 5.33 |
| KAT1 | 459 | 459 | 10 | 2.18 |
| KOUG | 565 | 460 | 105 | 18.58 |
| MELI | 668 | 623 | 45 | 6.74 |
| ULAB | 360 | 331 | 29 | 8.06 |
| URUM | 324 | 316 | 8 | 2.47 |
| YEL2 | 871 | 830 | 41 | 4.71 |

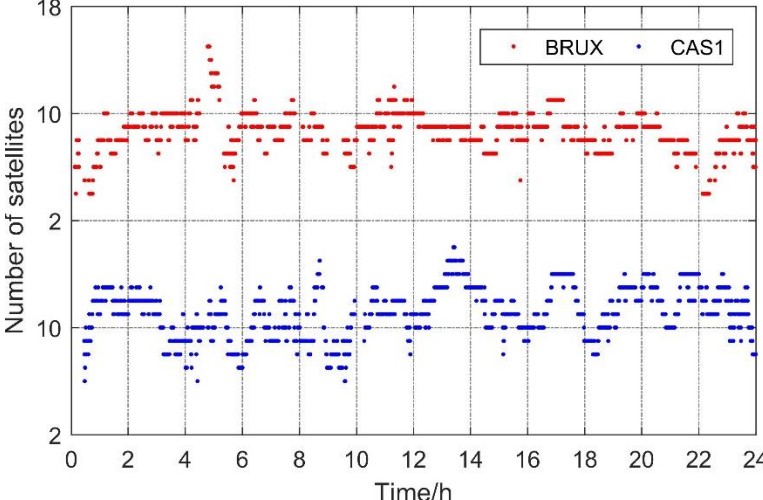

**Figure 4.** Number of satellites before narrow-lane ambiguity resolution.

For the kinematic PPP solution, the coordinate parameters of the station were estimated by epoch as white noise, which can better reflect the performance of the established PPP model [27]. As shown in Figure 5, the PPP solutions for the three different schemes were performed. Compared with the float PPP, the positioning accuracy in three directions significantly improved when the Var-sort method and the proposed method were used. Statistics show that when the Var-sort method was used, the average RMSs of kinematic PPP residual of all stations were approximately 1.55, 1.45 and 2.67 cm in the east, north and vertical direction, respectively, and the positioning accuracy in horizontal direction and vertical direction improved by approximately 42.5% and 36.3% compared with the float PPP solution. When the new proposed method was used, the average RMSs of the kinematic PPP residual were approximately 1.39, 1.31 and 2.42 cm, respectively; the positioning accuracy in the horizontal direction and vertical direction improved by approximately 9.6% and 9.1% compared with the Var-sort method. It can be seen that the ambiguity resolution has a significant impact on GPS kinematic PPP. Moreover, compared with the traditional partial ambiguity subset selection method, the improvement of kinematic positioning performance can be further improved when the new proposed method is used. It should be noted that in the calculation of coordinate residual RMS in this paper, the coordinate residual in the PPP convergence period was excluded, and the residual data were collected from 2 h.

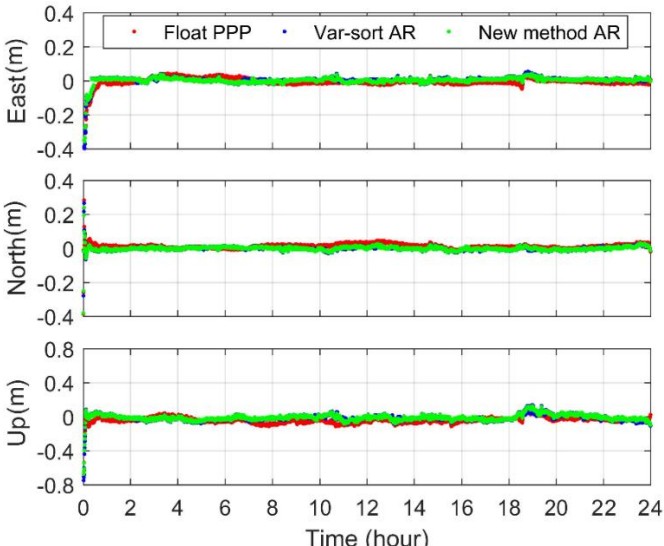

**Figure 5.** RMS of Kinematic PPP errors for three different directions at JFNG.

### 3.3. Discussion

The performance improvement of the fixed PPP solution is highly dependent on the quality control of the ambiguity resolution process. In order to objectively reflect the performance of the proposed optimal subset selection method on the positioning solution enhancement, it is necessary to demonstrate the ambiguity control strategy used in this study. A linear combination of wide-lane ambiguities was formed based on float ambiguity parameters at the original frequency. Based on its long wavelength characteristics, the study used a combination of the Bootstrapping success rate, the position accuracy information and the percentage of ambiguity resolution for judgment and strategy control. The wide-lane ambiguity resolution was determined based on whether the float ambiguity was close to an integer and Bootstrapping success rate after FCB correction [25]. For the narrow-lane ambiguity, due to the short wavelength, more strict quality control was needed, mainly including (1) starting ambiguity resolution when the position accuracy converges to the threshold value; (2) performing full ambiguity resolution first, and if it is unsuccessful, then partial ambiguity resolution is performed; and (3) using the combination of ambiguity success rates, ratio-test values and the number of ambiguity resolution for testing. The success rate threshold and ratio-test threshold were set as 0.99 and 0.25, respectively.

The number of satellites and PDOP used for narrow-lane ambiguity resolution is also different for different optimal subset selection strategies of partial ambiguity. In this paper, the CAS1 station was taken as an example. The ambiguity resolution state at 00:13:30 (UTC) using the two methods was extracted for analysis. The number of satellites available for narrow-lane ambiguity resolution at this station was 13. Firstly, the optimal subset of PAR was selected using the Var-sort method. The core of this method is to sequentially eliminate the satellites (pairs) with a large variance of narrow-lane float ambiguity until the ambiguity resolution condition is satisfied. According to this method, the ambiguity resolution condition was satisfied when four satellites were eliminated. The satellite sky distribution is shown in Figure 6a, with a PDOP value of about 1.9 and eight satellites. When the proposed optimal subset selection method was used, only one of the satellites needed to be eliminated. The satellite sky distribution is shown in Figure 6b, with a PDOP value of about 1.6 and 12 satellites. Therefore, in terms of both the number of satellites and the PDOP value for ambiguity resolution, the proposed method was better than the traditional method. This indicates that the spatial geometric configuration that consists of partial ambiguity subsets obtained by the new method was stronger, and the performance of the PPP fixed solution was also higher.

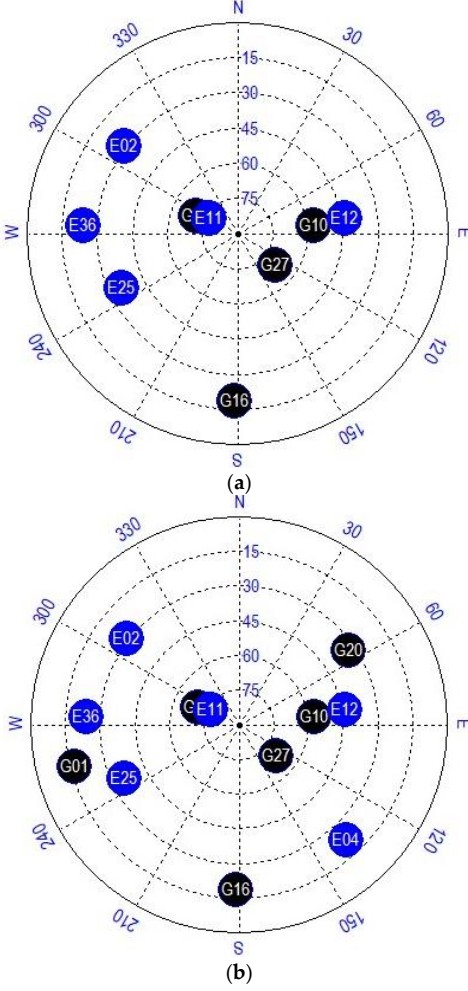

**Figure 6.** Skyplot of GPS/Galileo satellite at CAS1 station. (**a**) The traditional optimal subset selection method of PAR and (**b**) the proposed optimal subset selection method of PAR.

The GNSS observations used in the above experiments are obtained from the MGEX tracking station, which is generally located in a relatively empty observation environment. In the occurrence of no new satellite, cycle slip and signal interruption, i.e., when no satellite needs to be eliminated in the ambiguity resolution phase, the PDOP value does

not change significantly during the 24 h observation period. The ambiguity accuracy factor has a relatively large effect on ambiguity resolution in an open environment. Therefore, the weight factor of $c\_a$ and $c\_p$ were set as 0.3, 0.5 and 0.2, respectively. For the urban environment, due to the existence of a large number of tall buildings, tunnels, viaducts and trees, the GNSS signals are easily blocked, attenuated and produce multiple paths. These will seriously affect the fixation efficiency of GNSSS ambiguity and reduce the positioning reliability. To analyze the availability of the new method in an urban environment, the vehicle GNSS dynamic data of a Chinese city was selected. The test time was on the morning of 14 August 2020. After the observation period with severe occlusion before driving was removed, the data time of the sports car test was 3720s, and the sampling rate was 1s. The receiver type selects the SEPT ASTERX-M2 that can receive multi-system, multi-frequency GNSS data. The observation environment is a beltway with multiple viaducts and tunnels. Figure 7 shows the cycle slip, PDOP and the number of satellites tracking the observed data during the test period. It is worth noting that the satellite elevation angle is set as 10°, and the threshold of PDOP is set as 20 during the data observation quality analysis. Due to the influence of the viaduct and the tunnel, the number of satellites tracking changes greatly and rapidly, especially in the second half of the observation period. The corresponding PDOP is also constantly jumping around. In addition, the multiple paths caused by various reflected signals in the urban observation environment are also serious, which affect the solution accuracy of the float ambiguity parameter. Therefore, in this kind of observation environment, the strategy can appropriately improve the weight of the PDOP and ADOP factors when the optimal subset is selected with partial ambiguity resolution. The application results in this urban environment need to be further verified, and stricter quality control is needed when the ambiguity is fixed.

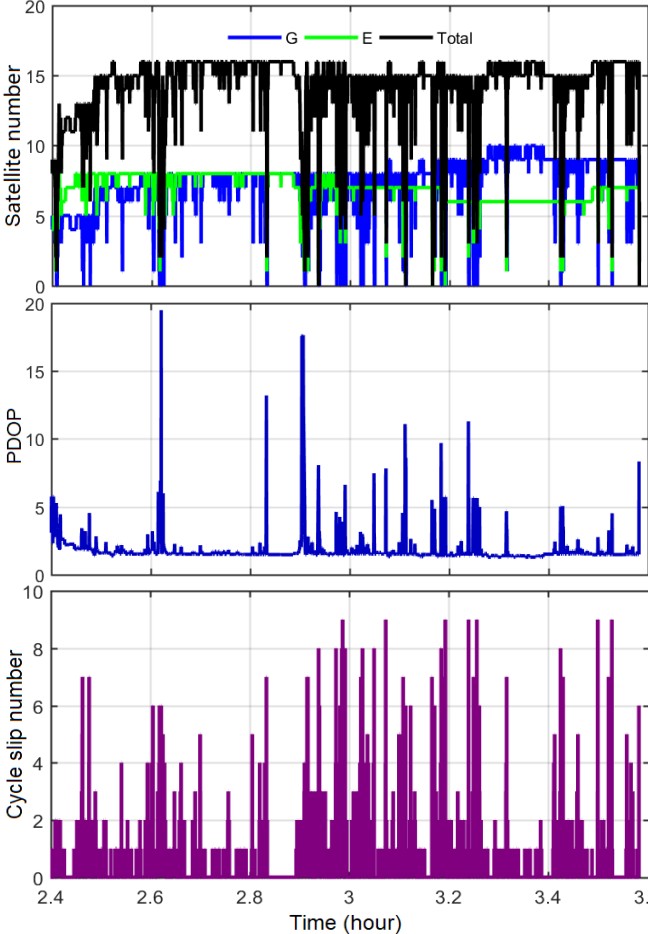

**Figure 7.** Number of satellites, PDOP and cycle slip tracking of vehicle dynamic observation data.

## 4. Conclusions

In order to avoid the adverse effects of full ambiguity resolution, many partial ambiguity resolution methods have been proposed and improved, thus improving the performance of the precision point positioning solution and the success rate of ambiguity resolution. The core of partial ambiguity resolution lies in the selection of the optimal subset. An inappropriate subset selection method will lead to a significant change in the geometric configuration of satellite positioning, thus affecting the accuracy of coordinate solutions. Therefore, in this study, a multi-factor constrained optimal subset selection method was proposed, which incorporated ambiguity variance, ADOP, satellite PDOP and ratio-test value. The detailed procedures to implement the proposed method were also demonstrated. In order to verify the feasibility of the proposed optimal subset selection method, the partial ambiguity resolution using the proposed method and the Var-sort method were performed based on the static observation data of 15 MGEX tracking stations. The results show that the proposed subset selection method can further improve the accuracy of the coordinate solution compared with the Var-sort method. The average root mean squares of the coordinate residuals decreased by about 12.90%, 6.83% and 9.39% in the eastern, northern and vertical directions, respectively. The number of epochs with ambiguity resolution increased by 0.87%~33.33%, with an average of 8.71%. According to the satellite PDOP values used for narrow-lane ambiguity resolution, the partial ambiguity subsets obtained using the proposed method induced a stronger spatial geometric configuration. The performance improvement of the PPP fixed solution was also more significant.

**Author Contributions:** All authors made significant contributions to this study. Conceptualization, C.Y. and S.X.; methodology, Y.D.; software, C.Y., S.G. and C.X.; validation, C.Y., H.W. and S.G.; formal analysis, C.Y.; writing—original draft preparation, C.Y.; writing—review and editing, S.G.; funding acquisition, C.Y. and Y.D. All authors have read and agreed to the published version of the manuscript.

**Funding:** This research was funded by the National Nature Science Foundation of China, grant number 41974010, Beijing Key Laboratory of Urban Spatial Information Engineering, grant number 20220120 and Shandong Province College Students innovation and Entrepreneurship Project, grant number CXCY2022088.

**Data Availability Statement:** The GPS precise ephemerides product provided by Wuhan University is available at ftp://igs.gnsswhu.cn, accessed on 30 December 2019. The MGEX observation for verifying the feasibility of the proposed optimal subset selection method is available at ftp://igs.ign.fr//pub/igs/data/, accessed on 28 December 2019. The precise coordinates of the MGEX stations are available in the file "igs20Pwwwwd.snx" released by IGS, respectively. All the data and products are publicly available through the respective organizations' website.

**Acknowledgments:** All authors gratefully acknowledge WHU and IGS for providing the data, orbit and clock products.

**Conflicts of Interest:** The authors declare no conflict of interest.

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
