# Peer review of "A New Optimal Subset Selection Method of Partial Ambiguity Resolution for Precise Point Positioning"

_remotesensing, doi:10.3390/rs14194819_

Round 1

Reviewer 1 Report

The article entitled " A new optimal subset selection method of partial ambiguity resolution for precise point positioning" presents a proposed a multi-factor constrained optimal subset selection method for the PAR. GPS/Galileo based on the static observation data of 15 Multi-GNSS Experiment (MGEX) tracking stations were used to verify the feasibility of two schemes PAR test. According to the authors, the proposed subset selection method improved the accuracy of the coordinate solution and the strength of geometric figure positioning when compared with the ambiguity variance sorting method. Results obtained for mean RMS of quantitative and qualitative results was made and discussed.

In my opinion, the problem is stated, the state of the art is presented, the solution and the data set with which the experiments were performed presented. The results are complemented with significant figures to help in results visualization. Overall, major parts of the manuscript are well written and organized. I find the results section very interesting and provides useful information for the readers. The conclusion subsection is well summarized. However, I have few recommendations for the authors to improve their manuscript in the methods section. First recommendation is related to flow of the methods section. I noticed few typos and unclear statements that need correction and clarification, respectively. Second, the discussion section demand revision. It fails to discuss the key message and recommendation of the key findings. Overall, the methods and discussion sections need improvement.

Author Response

Many thanks for giving us an opportunity to revise our manuscript. We appreciate your important and valuable comments. We have studied the reviewers' and editor's comments carefully and made revisions in the manuscript. A point-by-point response letter was also attached.

Please feel free to let us know if you have any questions.

Kind regards,

Caiya Yue

Responses to reviewers:

To Reviewer #1:

I have few recommendations for the authors to improve their manuscript in the methods section. First recommendation is related to flow of the methods section. I noticed few typos and unclear statements that need correction and clarification, respectively. Second, the discussion section demand revision. It fails to discuss the key message and recommendation of the key findings. Overall, the methods and discussion sections need improvement.

RESPONSE: Thank you for these suggestions. (Lines: 192-262; Lines: 473-506)

We have made careful modifications according to the comments.

We carefully examined and revised the method section. We focused on revising the typos and unclear statements. The modified part we have marked in red.

Thanks to the experts for suggested suggestions. We have added some new content in the discussion section. The partial ambiguity optimal subset selection method proposed in our manuscript may have better applicability to observations in complex environments. This complex observation environment mainly includes severe signal occlusion and multi-path interference. Therefore, the vehicle GNSS dynamic data of a Chinese city was selected to analyze the availability of the new method in urban environment. The test time was on the morning of August 14, 2020. After the observation period with severe occlusion before driving was removed. The data time of the sports car test was 3720s, and the sampling rate was 1s. Furthermore, we discussed selection strategies for three weight factors in open and urban environments. Regarding the ambiguity fixing performance in urban environments, we need to further verify it.

The main additions are as follows:

The GNSS observations used in the above experiments are obtained from the MGEX tracking station, which is generally located in a relatively empty observation environment. In the occurrence of no new satellite, cycle slip and signal interruption, i.e, when no satellite needs to be eliminated in the ambiguity resolution phase, the PDOP value does not change significantly during the 24-hour observation period. The ambiguity accuracy factor has a relatively large effect on ambiguity resolution in an open environment. Therefore, the weight factor of and ,  and  were set as 0.3,0.5 and 0.2, respectively. For the urban environment, due to the existence of a large number of tall buildings, tunnels, viaducts and trees, the GNSS signals are easy to be blocked, attenuated and produce multiple paths. These will seriously affect the fixation efficiency of GNSSS ambiguity and reduce the positioning reliability. To analyze the availability of the new method in an urban environment, the vehicle GNSS dynamic data of a Chinese city was selected. The test time was on the morning of August 14, 2020. After the observation period with severe occlusion before driving was removed, the data time of the sports car test was 3720s, and the sampling rate was 1s. The receiver type selects the SEPT ASTERX-M2 that can receive multi-system multi-frequency GNSS data. The observation environment is a beltway with multiple viaducts and tunnels. Figure 7 shows the cycle slip, PDOP and the number of satellite tracking of the observed data during the test period. It is worth noting that the satellite elevation angle is set as 10° and the threshold of PDOP is set as 20 during the data observation quality analysis. Due to the influence of the viaduct and the tunnel, the number of satellite tracking changes greatly and rapidly, especially in the second half of the observation period. The corresponding PDOP is also constantly jumping around. In addition, the multiple paths caused by various reflected signals in the urban observation environment are also serious, which affects the solution accuracy of the float ambiguity parameter. Therefore, in this kind of observation environment, the strategy can appropriately improve the weight of the PDOP and ADOP factors when the optimal subset is selected with partial ambiguity resolution. The application results in this urban environment need to be further verified, and stricter quality control is needed when the ambiguity is fixed.

Reviewer 2 Report

A multi-factor constrained optimal subset selection method for the PAR was proposed in the manuscript to improve the performance of PPP-AR. However, the theory and the experiments validation are insufficient.

1.         According to the description from lines 197-206, it is similar to the method of exhaustion. By combining the n-m selected good satellites with any combinations from the m candidate satellites, and select the fixing results. Then the step from 213 to218 seems unnecessary. Please clarify the significance of the proposed method.

2.         From Table.1, the accuracy of the static PPP is only 1cm-2cm. The results are relatively worse than that of the hourly PPP from the published paper (Table 4 in the Geng et al.) . Please describe the reason.

[1] Geng J, Yang S, Guo J. Assessing IGS GPS/Galileo/BDS-2/BDS-3 phase bias products with PRIDE PPP-AR[J]. Satellite Navigation, 2021, 2(1): 1-15.

3.         An optimal ambiguity subset selection method is the key for the PPP ambiguity resolution. The experiment is conducted based on the GPS/Galileo observations with the fixed m=4. However, the BDS phase biases products is already available for PPP-AR, why it is not used for the experiments?. Besides, the performance of the method should be validated in the kinematic PPP.

Author Response

Many thanks for giving us an opportunity to revise our manuscript. We appreciate your important and valuable comments. We have studied the reviewers' and editor's comments carefully and made revisions in the manuscript. A point-by-point response letter was also attached.

Please feel free to let us know if you have any questions.

Kind regards,

Caiya Yue

Responses to reviewers:

To Reviewer #2:

1) According to the description from lines 197-206, it is similar to the method of exhaustion. By combining the n-m selected good satellites with any combinations from the m candidate satellites, and select the fixing results. Then the step from 213 to218 seems unnecessary. Please clarify the significance of the proposed method.

RESPONSE: Thanks for these suggestions. (Lines: 229-231)

The first four steps of the partial ambiguity optimal subset selection method in this manuscript aim to enumerate all small subsets. In these subsets, there are multiple combinations satisfying the checking condition. So, starting from the fifth step, the main goal is to identify the optimal subset of all the candidates suitable for the current observation environment. We applied the method of normalizing the ratio-test, ADOP and PDOP values to dimensionless quantities between 0 and 1 for all combinations. At the same time, the weight factor is introduced to select the optimal subset to achieve efficient ambiguity resolution.

To help the reader to understand the process of this method, we added a sentence at the end of the fourth step. This sentence is to summarize the first four steps and guide the subsequent steps.

2) From Table.1, the accuracy of the static PPP is only 1cm-2cm. The results are relatively worse than that of the hourly PPP from the published paper (Table 4 in Geng et al.) . Please describe the reason.

[1] Geng J, Yang S, Guo J. Assessing IGS GPS/Galileo/BDS-2/BDS-3 phase bias products with PRIDE PPP-AR[J]. Satellite Navigation, 2021, 2(1): 1-15.

RESPONSE: Thanks for these suggestions.

We have carefully studied the paper of Geng et al. (2021). Table 4 presents the localization accuracy in RMS, whereas the coordinate residue is given directly in our manuscript. The two statistical methods differ, but both can be used to evaluate the positioning performance. For contrast purposes, we counted the RMS of the positioning residuals in our experiments, which were 0.78,0.66 and 1.37cm in the east, north and elevation directions, respectively, and basically in the same magnitude as the results of Geng et al. (2021).

Even if both parties are positioned with the GPS/Galileo system, the results are slightly different. According to our experience, the main reasons include benchmark products for positioning (such as precision ephemeris and clock offset), phase products for ambiguity resolution, data processing software, different GNSS stations and statistical methods.

Once again, thank the experts for this suggestion. We still need to further refine the model and improve the software for future data processing.

3) An optimal ambiguity subset selection method is the key for the PPP ambiguity resolution. The experiment is conducted based on the GPS/Galileo observations with the fixed m=4. However, the BDS phase biases products is already available for PPP-AR, why it is not used for the experiments?. Besides, the performance of the method should be validated in the kinematic PPP.

RESPONSE: Thanks for these suggestions. (Lines: 405-425)

We start by explaining why the Beidou system (BDS) was not used in our experiments. The first reason is that the Beidou-2 system (BDS-2) only serves the Asia-Pacific region, while in our experiments the global regional GNSS stations are used. The second reason is that the BDS-2 precision orbit and clock offset have relatively low accuracy and less stability. Therefore, we didn't use the BDS-2 satellite in the experiment. The third reason is the Beidou-3 system (BDS-3). Although the BDS-3 has now covered the whole world, the hardware delay deviation and data processing model of the BDS-3 are still not accurate enough, making it difficult to accurately separate the phase bias products required for the ambiguity resolution. For these reasons, we did not use the BDS-3 satellite in our experiments. However, we believe that when the BDS-3 satellite has enough high-precision phase delay bias correction products, the method presented in the manuscript would be further improved, and the positioning accuracy improvement effect will be more significant.

Following expert advice, we supplemented the kinematic PPP experiment. Results show that when the newly proposed method is used, the positioning accuracy in the horizontal direction and vertical direction improved by approximately 9.6% and 9.1% compared with those using the Var-sort method.

Reviewer 3 Report

The authors present a novel algorithm for PAR in PPP by

introducing a weighting function for choosing promising

ambiguity combinations before fixing. In general I agree with the content of the paper. There are some minor grammar issues and also a few technical questions which are summarized below

my comments refer to the line numbering scheme of the manuscript

1) l14

just re-phrase; hard to read

'.. of fixeed solution checking in full ambiguity..'

2) l40

I am not sue about the example in this reference paper [3], but I wonder about 'the reflectometry of precipitable water vapor'

usually soil moisture infers reflected GNSS signal .. while precipitable water just is the reasons for signal delay or in general signal refraction

check once more if the wording above is correct

3) l57

'.. due to the limited accuracy of the..'

4) l76/77

'.. ambiguity resolution (PAR) method. Many studies have ...'

5) l111-116

In the last paragraph of chapter 1 I recommend to use present tense instead of imperfect tense  , e.g.

'.. method of PAR is proposed. This method integrates ambiguity variance...'

'.. values and avoids the shortcoming..'

'.. method is verified...'

6)l 132

'where y is...'

7) l168

 '.. of float ambiguities in the subset...'

8) l182

'.. selection method is proposed.'

9) In equations (8) and (9) you specify the formulas for ADOP and PDOP

Although the formulas are correct the text is slightly inconsistent

as before you have denoted m as the subset of ambiguties;

In (8) and (9) n is specified as index of the subset, and later on the subset is again m; I see that you have to distinguish between the small subsets and the addition of n-m plus the small subset; but please check the indices for consistency to allow for an easy reading

10) l192

'where Q...'

11) l203/204

here you note 'The last m (m<n) satellites with larger variance..'

I would recommend to write instead 'The last m (m<n) ambiguities with larger...'

12) l206

explain in more detail to the reader whats the meaning of 'complexity of observation conditions ' ? 

do you mean low elevation ? or bad geometry? or ??

explain also in this respect the quantities 'C'

13) l 210

'.. the inverse of the ratio-test...'

14) l211

'.. inverse of the ratio-test...'

15) l235

'.. The location of these stations...'

16) l240

'.. Var-sort method) was chosen for...'

17) l242

'Then, the partial ambiguity resolution...'

18) Figure 2

enter in the legend which color refers to the float and which color to the fixed solutions

19) l286/287

'.. the optimal subset. Due to its long wavelength, the wide-lane ambiguity can be easily ...'

20) l301

'The coordinate residuals..' 1 blank too much

21) l 304

 '.. the horizontal and up directions...'

22) l313

'.. with a successful partial..'

23) l326

 what do you mean with 'and staying in the epochs'??

not understandable - re-phrase

24) l330

' I think the word 'regulated' is wrong here ?

moreover the quantities 4.68 and 8.11 might be means over epochs, but in general satellites are counted in integer numbers

25) l335/336

 'the variation characteristics of.. that the observation quality...'

 whats the exact meaning here? do you refer to the different number of visible satellites?

and moreover whats meant with observation quality? : just the number of satellites -or- geometry- or any other indicator?

26) l339

'.. partial ambiguity offer a more significant improvement in the scenarios..'

27) l342/343

 is the word 'combining' here optimal ?

maybe: ' by evaluating the coordinate residuals and ambiguity resolution rate in parallel.'

28) Table 2, columns 'Var-sort' and 'Proposed Method'

Did you mix up the content of these 2 columns? In my understanding

the quantities in column 'Proposed method' should be larger than in

'Var-sort' method. But its just the other way round ?

29) l359

'.. wide-lane ambiguties was formed...'

30) l364

'.. For the narrow-lane...'

31) l407

'.. and the Var-sort method were performed...'

32) l411

'The number of epochs with ambiguity resolution...'

best regards

Author Response

Many thanks for giving us an opportunity to revise our manuscript. We appreciate your important and valuable comments. We have studied the reviewers' and editor's comments carefully and made revisions in the manuscript. A point-by-point response letter was also attached.

Please feel free to let us know if you have any questions.

Kind regards,

Caiya Yue

Responses to reviewers:

To Reviewer #3:

We would first like to particularly thank the experts for so many detailed comments. We believe that your suggestions will be very helpful to improve the quality of the manuscript. We have carefully revised your suggestions. Here, we have targeted responses to several important revision suggestions.

1) L40. I am not sure about the example in this reference paper [3], but I wonder about 'the reflectometry of precipitable water vapor'. Usually soil moisture infers reflected GNSS signal. while precipitable water just is the reasons for signal delay or in general signal refraction. Check once more if the wording above is correct.

RESPONSE: Thanks for these suggestions. (Line: 40; Lines: 552-553)

We found a more appropriate reference to replace the literature in the manuscript. The delay of the GNSS signal when crossing the troposphere can be divided into two parts, i.e., dry delay and wet delay. The wet delay is the main data source for water vapor reduction in the atmosphere. This is a simple process of inversion of precipitable water vapor with GNSS.

2) In equations (8) and (9) you specify the formulas for ADOP and PDOP. Although the formulas are correct the text is slightly inconsistent as before you have denoted m as the subset of ambiguties; In (8) and (9) n is specified as index of the subset, and later on the subset is again m; I see that you have to distinguish between the small subsets and the addition of n-m plus the small subset; but please check the indices for consistency to allow for an easy reading

RESPONSE: Thanks for these suggestions. (Lines:207-211; Lines: 219-225)

We have reorganized and revised this section.

3) L206. explain in more detail to the reader whats the meaning of 'complexity of observation conditions ' ? do you mean low elevation ? or bad geometry? or ??explain also in this respect the quantities 'C'.

RESPONSE: Thanks for these suggestions. (Lines: 222-225)

We have modified this section. In this manuscript, the complex condition mainly refers to the degree of signal occlusion and multipath interference around the GNSS station.

4) Figure 2. enter in the legend which color refers to the float and which color to the fixed solutions.

RESPONSE: Thanks for these suggestions. (Lines: 315-317)

We have modified this figure.

Figure 2. Coordinate residual series of float and fixed solutions at BOR1 station

5) L335/336. The variation characteristics of.. that the observation quality...' whats the exact meaning here? do you refer to the different number of visible satellites?and moreover whats meant with observation quality? : just the number of satellites -or- geometry- or any other indicator?

RESPONSE: Thanks for these suggestions.

For this study, the quality of observations here refers to the number of satellites. The number of satellites affects the satellite geometry figure, especially in some areas of severe occlusion.

Once again, thanks the reviewer for your suggestions.

Round 2

Reviewer 2 Report

The author has clearly reply the questions and it is improved.